# Digital Health Literacy and Information-Seeking in the Era of COVID-19: Gender Differences Emerged from a Florentine University Experience

**DOI:** 10.3390/ijerph20032611

**Published:** 2023-01-31

**Authors:** Guglielmo Bonaccorsi, Veronica Gallinoro, Andrea Guida, Chiara Morittu, Valerio Ferro Allodola, Vieri Lastrucci, Patrizio Zanobini, Orkan Okan, Kevin Dadaczynski, Chiara Lorini

**Affiliations:** 1Department of Health Science, University of Florence, Morgagni Blvd 48, 50134 Florence, Italy; 2Health Literacy Laboratory, Department of Health Science, University of Florence, Morgagni Blvd 48, 50134 Florence, Italy; 3School of Specialization in Public Health, University of Florence, Morgagni Blvd 48, 50134 Florence, Italy; 4Epidemiology Unit, Meyer Children’s University Hospital, 50139 Florence, Italy; 5Department of Sport and Health Sciences, Technical University Munich, Uptown München-Campus D, Georg-Brauchle-Ring 60/62, 80092 Munich, Germany; 6Department of Health Sciences, Fulda University of Applied Sciences, 36037 Fulda, Germany; 7Center for Applied Health Science, Leuphana University Lueneburg, 21335 Lueneburg, Germany

**Keywords:** gender differences, COVID-19, online health information seeking behaviour, Digital Health Literacy

## Abstract

Gender appears to be a strong predictor of online health information-seeking behaviour (OHISB), which is related to Digital Health Literacy (DHL). Gender differences in OHISB have been studied in different countries with different results, but no studies have investigated gender-specific OHISB among University students during the COVID-19 pandemic. We sought to investigate any gender differences in OHISB in the period between the first and second waves of the pandemic in Italian university students. A questionnaire developed by the global COVID-HL network, including existing and adapted validated scales and self-developed scales, was administered to 2996 University students in Florence. Gender differences were tested using the χ^2^ test or the Mann–Whitney U test. Male students reported a higher score in DHL than females (*p* < 0.001). However, female students seek COVID-19 information more often on different sources (for themselves and other people), on various topics, consider various aspects of information quality to be “very important’’ (*p* < 0.05) and are more likely to be “often dissatisfied’’ or ”partly satisfied’’ with information (*p* < 0.001). Our study confirmed gender as an important dimension to explain students’ OHISB differences, which could help institutions promote gender-specific education programmes and provide gender-oriented health information.

## 1. Introduction

Access to health information is nowadays getting easier thanks to the Internet. In January 2021, almost 60% of the global population were active internet users [1]. According to a survey, 55% of the European population has been searching the Internet for health information, with higher percentages in Germany and Denmark [2]. Everyone can share and post different information online without verification or support by scientific evidence, leading to an overwhelming amount of true, false and mixed news provided by different digital communication channels [3]. This can result in the “infodemic”, defined by the World Health Organization (WHO) in 2020 as ”too much information including false or misleading information in digital and physical environments during a disease outbreak’’. The spread of bad information can be incorrect or useless (“misinformation”) or deliberately twisted to justify a political, ideological or other doctrinaire position (“disinformation”).

The way people look for information regarding health-related topics has been defined as ‘‘Health Information-Seeking Behaviour’’ (HISB) [4] and, specifically for online search, Online health information-seeking behaviour (OHISB). Online HISB is affected by many sociodemographic factors. Previous studies have in fact reported that adults with a lower socioeconomic status, male, older, with lower education levels, and poor Internet connectivity were less likely to look for online health information [5,6,7]. Online health information seeking is also affected by personal health status [3]. People who lack adequate health literacy—and, specifically, digital health literacy (DHL)—could have problems finding appropriate information, as well as navigating the digital ocean of news generated by the infodemic [8].

DHL is defined as the “ability to seek, find, understand and appraise health information from electronic sources and apply the knowledge gained to addressing or solving a health problem” [3]. Lower DHL is associated with poor health behaviour and worse self-perceived understanding of health status, symptoms and optional treatments [9]. In the era of COVID-19, higher digital health literacy, higher frequency of searching for symptoms of SARS-CoV-2 infection, higher perceived importance of quickly learning from the information searched, and lower frequency of searching on topics regarding psychological stress were found to be predictors of information satisfaction [3]. DHL is a key competence to navigate web-based COVID-19-related information and service environments and may play a key role in adherence to COVID-19 prevention behaviours [9,10,11].

Those with higher levels of information satisfaction also reported a higher importance of whether the information was up to date, verified or came from official sources, as well as whether the reader could quickly learn the most important things, and the content was understandable [3]. Information satisfaction is, however, of little use if it does not guide individuals in making relevant and appropriate health decisions [12].

Moreover, the research by Swire et al. showed, in this regard, that people prefer information that confirms their pre-existing attitudes (i.e., selective exposure). Previous beliefs and ideologies could impede checking the facts contained in a false report, although fact checking could also be counterproductive in certain circumstances [13]. Indeed, research about fluidity and familiarity bias in political debates shows that people tend to remember information based on their opinions rather than the context in which they receive it. Furthermore, people are more likely to accept “familiar” information as true, with the consequent risk that the repetition of false information in the context of fact-checking could increase the likelihood of accepting them as true [14].

Health decisions and, more generally, health behaviours taken by people are affected by gender. Gender-specific literature provides mixed evidence regarding the relationship with health literacy: in some studies [15,16], females tend to have higher levels of health literacy, while in others there are no gender differences except for specific aspects—for instance, “Social support for health” [17]. In contrast, health literacy (HL) was found to be higher in males in a Chinese study [18]. 

Gender differences in Internet use have been studied in many countries, with different results [19,20]. Regarding HISB, many studies show that women are more engaged in searching for health information in general and on the Internet in particular: in fact, being a woman is one of the strongest predictors of OHISB [21,22].

To the best of our knowledge, no studies have investigated gender differences in OHISB among university students during the COVID-19 pandemic, although this could be relevant to achieving effective public health interventions to mitigate the negative effects of the pandemic. For this reason, this study aims to explore any gender differences in OHISB among students at the University of Florence during the timeframe between the first and second pandemic waves of SARS-CoV-2 circulation in Italy. 

## 2. Materials and Methods

### 2.1. Study Design

Data were collected using a questionnaire developed by Dadaczynski and colleagues [23] and developed within the COVID-Health Literacy network). (www.covid-hl.org) (access on: 7 June 2022). In Florence, the COVID-HL University Students Survey was conducted by sharing the online questionnaire with the students of all the study courses (bachelor’s, master’s, PhD, Postgraduate School) via institutional email. In addition, the survey was advertised on the University of Florence web pages and social networks. There were no exclusion criteria apart from not being a Florence University student.

In 2020, the number of students attending the University of Florence was about 50,000. The questionnaires were administered from 17 August to 3 October 2020, and students’ participation was voluntary. All the data were processed anonymously and not attributable to a specific person, in accordance with European Regulation 2016/679 and Italian Legislative Decree 101/2018.

### 2.2. Questionnaire

The questionnaire included either existing validated scales adapted to the COVID-19 pandemic or newly developed scales [23]. 

The original questionnaire developed in the English language was translated into Italian by using a standard procedure [24]: two independent native Italian speakers translated the questionnaire into Italian, and then two independent native English speakers backtranslated the two versions into English. The four versions (two in Italian and two in English) were assessed and discussed by the research group to verify any discrepancies emerging from the process and then a shared final version was drawn up. The Italian version of the COVID-19 adapted version of the DHLI has recently been validated [25].

For the purpose of this study, the following sections of the questionnaire were included in the analysis: sociodemographic characteristics (sex, age, country of origin, study course),Digital Health Literacy (DHL),OHISB, in particular: self-versus surrogate seeking, sources used for online information seeking, Corona-related topics searched, importance of internet information search, information satisfaction.

Self-versus surrogate-seeking was investigated with the question “Have you searched the Internet in the last four weeks for information about the coronavirus?’’ with the following response options: (1) yes, only information for me, (2) yes, only information for other people, (3) yes, information for me and other people, (4) no, I haven’t searched for information in the last four weeks.

To measure the COVID-19 DHL, the Digital Health Literacy Instrument (DHLI) developed by Van der Vaart and Drossaert [9] was used, adapting questions to coronavirus instead of general health. It included five subscales (information searching, adding self-generated content, evaluating reliability, determining relevance, protecting privacy). Each subscale included three items that could be rated with four possible responses (“very difficult, difficult, easy, very easy,” except for the item ‘’Protecting privacy,’’ where the response options were “never, once, several times, often”). A total scale score was calculated as the mean value of the scores for each item, excluding the “Protecting privacy’’ items due to low validity reported in the validation study [25].

Sources used for online information seeking were assessed through a specific question (“Now various possibilities are mentioned to get information about the coronavirus and related topics on the Internet. Please indicate how often you currently use these sources”) adapted from a study by the Bertelsmann Foundation [26], with five possible responses: (1) often, (2) sometimes, (3) rarely, (4) never, and (5) don’t know.

To investigate corona-related topics searched, a specific question (“Please indicate the specific topics you are searching for in the context of the coronavirus”) was self-developed, including ten items that could be answered with “no” or “yes”.

The importance of Internet information search was investigated through the question, “Now it’s about how important various things are to you when you search the Internet for coronavirus and related topics. How important is it to you that…” by Gebel, Juenger and Wagner [27], with six items and the following response options: (1) not at all important, (2) rather not important, (3) rather important, (4) very important.

Lastly, information satisfaction was measured through a self-developed question (“How satisfied are you with the information you find on the Internet about the coronavirus?”) and five response options: (1) very dissatisfied, (2) dissatisfied, (3) partly satisfied, (4) satisfied, and (5) very satisfied.

### 2.3. Statistical Analysis

Normality of continuous variables was assessed using the Kolmogorov–Smirnov test. Continuous variables were described using mean and standard deviation (SD), or median and interquartile range (IQR), as appropriate. Categorical variables were expressed as percentages. The association between gender and the other investigated variables was tested using the Chi2 test. The difference in Digital Health Literacy between genders was evaluated using the Mann–Whitney U test. For all the analyses, the alpha level was considered as significant at *p* < 0.05. The analyses were performed using IBM SPSS 28.0 (IBM Corp., Armony, NY, USA).

## 3. Results

### 3.1. Descriptive Analysis of the Sample

As a whole, a convenience sample of 2996 university students participated in the study, of whom 68% were female. The median age was 22 (IQR: 20–24; range: 18–70 years).

Ninety-two percent were born in Italy, 1.7% in Albania, and 5.8% in other countries (China, Romania, Poland, Cameroon, Russia). Fifteen percent attended a study course in the Medical or Health Sciences area, 13% in Engineering, 11% in Humanities, 10% in Economics/Statistics, 9% in Architecture/Urban and Environmental Sciences, and 5% in Education Sciences, while the remaining 37% were engaged in other disciplines. Most of the students (62.1%, *N* = 1862; 65.5% among males; 60.3% among females) attended a bachelor’s degree programme, 37.1% (*N* = 1111; 33.9% among males; 38.6% among females) attended a master’s degree course, 0.8% (0.6 among males; 0.8% among females) a PhD or post-doc programme. 

### 3.2. Online COVID-19 Information-Seeking Behaviour by Gender

#### 3.2.1. Self-Versus Surrogate Seeking

The majority of the sample reported having searched the internet in the last four weeks for their own purposes and that of other people (50.9%), with a statistically significant association with gender (*p* < 0.001). In particular, among males, 28.6% searched for information only for themselves, 2.8% only for other people, and 45% for themselves and for other people; among women, 19% searched for information only for themselves, 2.4% only for other people, and 53.7% searched for both. Almost 25% (23.6% of men and 24.9%) of the sample reported having not searched the Internet for corona-related information in the previous four weeks.

#### 3.2.2. Sources Used for Online Information Seeking

Regarding “Sources used for online information seeking”, ten areas were investigated (Table 1). Seven areas—websites of public bodies, social media, YouTube, blogs on health topics, health portals, websites of doctors or health insurance companies, news portals—showed a statistically significant association with gender (*p* < 0.05). The percentage of females using “often” one of these sources was always higher than the percentage of males, except for the use of “YouTube”. On the contrary, the ‘’rarely’’ option was chosen more frequently by males for all the sources except “YouTube”.

#### 3.2.3. Corona-Related Topics Searched

Table 2 shows the data for “Corona-related topics searched”. Ten items were investigated, five of which showed statistically significant differences between genders (*p* < 0.05). In particular, among females, the most searched topic concerned symptoms of COVID-19 disease, whereas men mainly sought information on the socioeconomic consequences of the coronavirus. Unsearched topics included “symptoms of COVID-19 disease” and “coping with mental stress caused by coronavirus” among men, while more women did not search for the socioeconomic impact of the disease.

#### 3.2.4. Importance of Internet Information Search

Table 3 shows the data relating to the “Importance of Internet information search”. All items investigated resulted in statistically significant gender differences (*p* < 0.05). Compared to male respondents, female students rated all items more often as “very important”, while the opposite holds true for the response option “not at all important” that could be more often observed for male students. Five out of six items were ‘’rather important’’ more for males than for females, except for the item “Important that different opinions are represented”.

#### 3.2.5. Satisfaction with COVID-19-Related Information

Data relating to satisfaction with COVID-19-related information showed a significant association with gender (*p* < 0.001). In particular, males more often consider themselves very dissatisfied (4.3% M; 2.4% F), satisfied (35% M; 22.9% F) and very satisfied (2.4% M; 1% F); on the contrary, women are more often dissatisfied (6.9% M; 7.2% F) or partly (51.4% M; 66.5% F).

## 4. Digital Health Literacy

Considering the mean score reported in the DHLI, male (2.98 ± 0.48, median value 3, IQR 2.67–3.33) students showed a statistically significant higher score than female (2.82 ± 0.47, median value 2.83, IQR 2.50–3.08), as shown in Figure 1 (*p* < 0.001).

## 5. Discussion

In our study, gender differences were found in OHISB. In fact, female students reported using more often different sources for online information seeking (except for YouTube), searching for more corona-related topics (except for economic and social consequences) and considering each item in the “Importance of internet information search” section as “very important”. Furthermore, female students are more likely to be “often dissatisfied” or “partly satisfied” with information about COVID-19 and to search more often for information for themselves and other people. Moreover, male students reported a higher score in DHL than did females.

The profile that we described confirms what other studies from the COVID-HL network have already found; in particular, in the German, Portuguese and Slovenian cohorts, female students showed a lower DHL [28,29,30]. Moreover, in the German cohort, women were less satisfied than men with information about COVID-19. Nevertheless, in other countries such as the US, China, Philippines and Pakistan, a higher score of DHL was found in females [12,31,32].

If we consider the general population and not only university students, many studies have reported differences in OHISB between genders. While women reported being more interested in health information and showed more active search activities [33], men were less likely to read health information [34]. This gender gap in OHISB has been found to be stable over time [5,35], and it might depend on different motivations for seeking health information, partly determined by traditional constructions of femininity and masculinity.

From a sociological point of view, masculine norms of behaviour tend to emphasise self-reliance, determination and emotional control, while feminine norms promote emotional sensitivity, compassion, caring and support activities [36,37]. As a result, when dealing with health problems, men may feel particularly uncomfortable discussing highly emotional and sensitive issues within their larger social networks. Conversely, women who are more accustomed to sharing personal experiences would feel comfortable searching for support from a wider network of individuals.

In addition, the search for help, whatever it may be, essentially implies the need to rely on others for assistance, which can be seen by men as a kind of personal weakness, contradicting the characteristics typically associated with masculinity, such as independence and self-confidence [38].

This is consistent with another recent finding that women were more likely than men to rely on OHISB for social motives and enjoyment [39]. OHISB reflects a need—especially among men—for health information that is clearly explained and tailored to their specific necessities [40].

As for the online sources explored, some studies [39,40] found that women used more health forums, blogs and search engines while men preferred to use apps and web-based encyclopaedias [28], though others found gender similarities in the use of health-related apps. Likewise, a Finnish study found that women prefer to search for medical information through multiple sources compared to men, not just online, but also from printed materials such as patient information leaflets, newspapers, books and magazines [40]. Another study found that a higher percentage of men claimed to seek health information on the Internet for their easiest accessibility [41].

In terms of topics searched, the literature affirms that men focus more on health policies and systems, health insurance companies and non-commercial health organisations [40]; this behaviour is confirmed also in the COVID-19 pandemic time, during which they searched information on the economic and social consequences of the COVID-19 pandemic more frequently than women [28].

Despite these mixed and contrasting results, which can be attributable to geographical, as well as socio-cultural or educational differences, gender should be considered a source of health literacy disparities, and it must be assessed in any interventions aimed at increasing HL.

Even though men and women reported equal access to online health information, OHISB should be explained using gender-specific models. Different concepts have been introduced to better explain these differences. In particular, recent research has indicated two distinct theories [40]. The biological one includes evolutionary theory, hormone exposure of the brain, and the selectivity hypothesis, as described by Meyers-Levy and Loken [42]. The second theory, sociocultural theory, affirms that women are more involved in activities such as staying in contact with family members because of their higher social engagement; on the other hand, men are more task oriented [43]. In light of this principle, the increased search for health information might be explained by gender-specific health-related tasks, such as care for children or elderly family members [44].

Another potential explanation for the differences between genders in the number of sources used for online health information and in information satisfaction could be found in the cognitive-behavioural model. According to this, there is a positive association between information-seeking and health anxiety, which may be generated by catastrophic cognitions. Since the COVID-19 pandemic has actually represented a global health catastrophe [45], we can assume that, among females, catastrophic cognitions are more frequent.

COVID-19 has impacted the lives of hundreds of millions of people, causing higher levels of distress in women than in men [46,47]. In fact, women with higher levels of anxiety are more likely to draw COVID-19 information from social networks, such as Facebook and Twitter [48]. Additionally, health anxiety is an issue of concern during the COVID-19 pandemic. Individuals with higher levels of health anxiety tend to search for health-related information to alleviate this [49]. While seeking information as an act of reassurance may temporarily alleviate one’s anxiety, overall, it will increase long-term anxiety due to negative reinforcement of the information-seeking behaviours [50]. This abnormal behavioural pattern, in which excessive or repeated online searches for health-related information are distressing or anxiety-provoking, is called cyberchondria [51]. Therefore, given the positive association between information seeking and health anxiety, assessing and reducing the frequency of information seeking may be helpful in reducing health anxiety. So, in the amidst of pandemic and infodemic, providing timely, relevant and accurate information, promoting the use of reliable information sources, and improving DHL to access, comprehend and appraise online information seems to be essential for physical and psychological well-being [52].

Apart from any possible explanation, gender differences in OHISB have to be considered in health communication by offering information according to male and female specific preferences, attitudes and behaviour. This does not mean that institutions have to fit the information by gender but that they have to adapt the same information to different sources, taking into account the gender-specific DHL and OHISB. Moreover, health information must be complete, accurate and truthful to prevent further anxiety-generating information-seeking behaviours. In fact, a key contributing factor related to cyberchondria is the ambiguity of online health information, such that it is often inaccurate, misleading or incomplete [53]. This could result in spending more time evaluating the validity of health-related information [50].

Our study has several strengths and limitations. All Florence University students were invited to join the survey, but only about 3,000 students decided to participate; no data on non-respondents are available, so selection bias cannot be excluded, which limits the generalisability of the results. Finally, the cross-sectional study design does not allow us to interpret the associations as causal. Moreover, the questionnaire was self-administered, so social desirability cannot be excluded; however, the survey was anonymous, and this may have limited such bias. Regarding its strengths, this study was part of an international research network (www.covid-hl.org) with a shared methodology and questionnaire, so robust comparisons can be made.

## 6. Conclusions

Gender differences in OHISB have to be considered in health communication by offering information according to male- and female-specific behaviour. In fact, female students have proven to have a lower DHL, to be less satisfied with the information, to use different research sources and to consider very important different facets of information. On the other hand, males seem to adopt more heuristic behaviours.

In this specific context, universities and high schools could and should support all their students, both female and male, in the process of appropriate health information seeking by, for example, promoting digital (health) literacy, helping them to better choose their information sources, and creating reliable and gender-specific websites or apps; in other words, they must support students’ awareness in the right way to be informed so as to prevent (mental) health problems. Improving the DHL is one of the main pillars to combat the infodemic, which in turn could prevent the undesirable consequences of dis- and misinformation.

## Figures and Tables

**Figure 1 ijerph-20-02611-f001:**
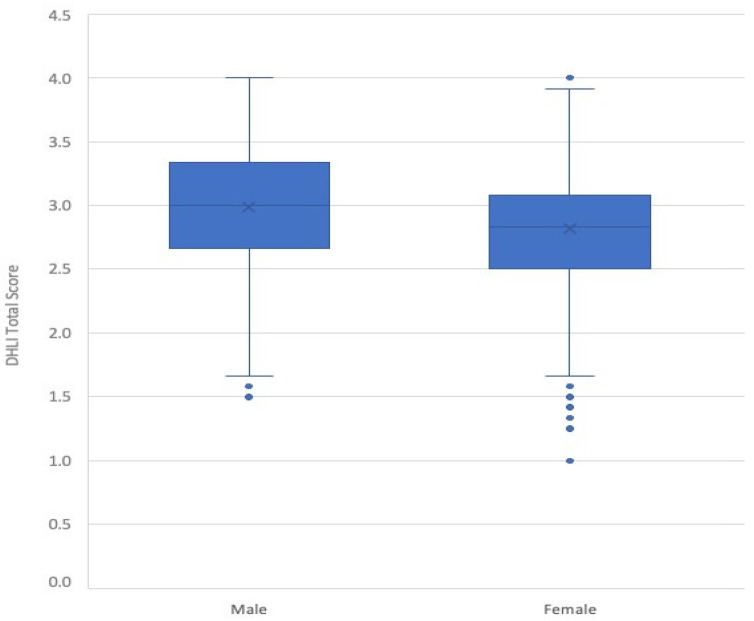
Box plot of the DHLI total score by gender (*N* = 2996).

**Table 1 ijerph-20-02611-t001:** Descriptive analysis of the sources used for online information seeking (*N* = 2996).

Sources Used for Online Information Seeking	Sex	Response Options										*p*-Value *
		Often		Sometimes		Rarely		Never		Don’t Know		
		N	%	N	%	N	%	N	%	N	%	
Search engines	M	693	73.9	174	18.6	51	5.4	16	1.7	4	0.4	0.919
	F	1502	75.0	350	17.5	102	5.1	39	1.9	10	0.5	
	ALL	2195	74.6	524	17.8	153	5.2	55	1.9	14	0.5	
Websites of public bodies	M	372	39.7	346	37.0	153	16.3	59	6.3	6	0.6	<0.001
	F	1033	51.7	638	31.9	257	12.9	63	3.2	6	0.3	
	ALL	1405	47.9	984	33.5	410	14.0	122	4.2	12	0.4	
Wikipedia and other online-encyclopedias	M	248	26.6	316	33.9	211	22.6	146	15.6	12	1.3	0.355
	F	549	27.5	607	30.4	503	25.2	308	15.4	27	1.4	
	ALL	797	27.2	923	31.5	714	24.4	454	15.5	39	1.3	
Social media (eg, Facebook, Instagram, Twitter)	M	215	23.1	201	21.6	235	25.2	257	27.6	23	2.5	<0.001
	F	682	34.1	494	24.7	449	22.5	336	16.8	39	2.0	
	ALL	897	30.6	695	23.7	684	23.3	593	20.2	62	2.1	
YouTube	M	174	18.7	184	19.8	255	27.4	288	31.0	28	3.0	0.032
	F	301	15.1	372	18.6	543	27.2	705	35.3	78	3.9	
	ALL	475	16.2	556	19.0	798	27.3	993	33.9	106	3.6	
Blogs on health topics	M	37	4.0	165	17.9	285	31.0	380	41.3	53	5.8	<0.001
	F	176	8.9	390	19.7	667	33.6	661	33.3	89	4.5	
	ALL	213	7.3	555	19.1	952	32.3	1041	35.9	142	4.9	
Online communities	M	39	4.2	94	10.2	233	25.2	487	52.8	70	7.6	0.259
	F	84	4.2	241	12.1	549	27.6	973	48.8	145	7.3	
	ALL	123	4.2	335	11.5	782	26.8	1460	50.1	215	7.4	
Health portals	M	151	16.2	315	33.8	232	24.9	195	20.9	38	4.1	<0.001
	F	527	26.5	663	33.3	478	24.0	266	13.4	56	2.8	
	ALL	678	23.2	978	33.5	710	24.3	461	15.8	94	3.2	
Websites of doctors or health insurance companies	M	48	5.2	169	18.2	251	27.0	392	42.2	70	7.5	<0.001
	F	192	9.6	412	20.7	578	29.0	660	33.1	153	7.7	
	ALL	240	8.2	581	19.9	829	28.3	1052	36.0	223	7.6	
News portals	M	271	29.1	361	38.7	174	18.7	102	10.9	24	2.6	<0.001
	F	739	37.0	746	37.3	346	17.3	143	7.2	25	1.3	
	ALL	1010	34.5	1107	37.8	520	17.7	245	8.4	49	1.7	

* Pearson’s chi-squared test.

**Table 2 ijerph-20-02611-t002:** Descriptive analysis of the corona-related topics searched (*N* = 2996).

Corona-Related Topics Searched	Sex	Response Options				*p*-Value *
		Yes		No		
		N	%	N	%	
Current spread of the coronavirus	M	838	87.4	121	12.6	0.702
	F	1790	87.9	247	12.1	
	ALL	2628	87.7	368	12.3	
Transmission routes of the coronavirus	M	300	31.3	659	68.7	0.336
	F	602	29.6	1435	70.4	
	ALL	902	30.1	2094	69.9	
Symptoms of the disease COVID-19	M	450	46.9	509	53.1	<0.001
	F	1121	55.0	916	45.0	
	ALL	1571	52.4	1425	47.6	
Individual measures to protect against infection	M	339	35.3	620	64.7	0.815
	F	729	35.8	1308	64.2	
	ALL	1068	35.6	1928	64.4	
Hygiene regulations	M	283	29.5	676	70.5	0.204
	F	648	31.8	1389	68.2	
	ALL	931	31.1	2065	68.9	
Current situation assessments and recommendations	M	434	45.3	525	54.7	0.029
	F	1009	49.5	1028	50.5	
	ALL	1443	48.2	1553	51.8	
Restrictions	M	615	64.1	344	35.9	0.03
	F	1388	68.1	649	31.9	
	ALL	2003	66.9	993	33.1	
Economic and social consequences of the coronavirus	M	487	50.8	472	49.2	<0.001
	F	819	40.2	1218	59.8	
	ALL	1306	43.6	1690	56.4	
Dealing with psychological stress caused by the coronavirus	M	132	13.8	827	86.2	<0.001
	F	466	22.9	1571	77.1	
	ALL	598	20.0	2398	80.0	
Other topics	M	35	3.6	924	96.4	0.424
	F	63	3.1	1974	96.9	
	ALL	98	3.3	2898	96.7	

* Pearson’s chi-squared test.

**Table 3 ijerph-20-02611-t003:** Descriptive analysis of the importance of an Internet information search (*N* = 2996).

Importance of Internet Information Search	Sex	Response Options								*p*-Value *
		Not at All Important		Rather Not Important		Rather Important		Very Important		
		N	%	N	%	N	%	N	%	
Important that information is up to date	M	15	1.6	0	0.0	139	14.5	805	83.9	<0.001
	F	16	0.8	2	0.1	171	8.4	1848	90.7	
	ALL	31	1.0	2	0.1	310	10.3	2653	88.6	
Important that information is verified	M	10	1	0	0	95	9.9	854	89.1	<0.001
	F	18	0.9	2	0.1	116	5.7	1901	93.3	
	ALL	28	0.9	2	0.1	211	7	2755	92	
Important to quickly learn the most important things	M	111	11.6	3	0.3	402	41.9	443	46.2	<0.001
	F	107	5.3	4	0.2	713	35	1213	59.5	
	ALL	218	7.3	7	0.2	1115	37.2	1656	55.3	
Important that information comes from official sources	M	23	2.4	2	0.2	135	14.1	799	83.3	<0.001
	F	23	1.1	4	0.2	166	8.1	1844	90.5	
	ALL	46	1.5	6	0.2	301	10	2643	88.2	
Important that different opinions are represented	M	184	19.2	35	3.6	437	45.6	303	31.6	0.035
	F	341	16.7	45	2.2	983	48.3	668	32.8	
	ALL	525	17.5	80	2.7	1420	47.4	971	32.4	
Important that the subject is dealt with comprehensively	M	43	4.5	5	0.5	258	26.9	653	68.1	<0.001
	F	56	2.7	9	0.4	419	20.6	1553	76.2	
	ALL	99	3.3	14	0.5	677	22.6	2206	73.6	

* Pearson’s chi-squared test.

## Data Availability

Data are available by request to the corresponding author.

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
