# Peer review of "Digital Health Literacy and Information-Seeking in the Era of COVID-19: Gender Differences Emerged from a Florentine University Experience"

_ijerph, 2023, doi:10.3390/ijerph20032611_

Round 1

Reviewer 1 Report

The paper claims the following : 

1) Authors claimed in line number 18 - 20 that no studies investigated gender-specific OHISB among University students during the COVID-19 pandemic.

2) A questionnaire developed by the global COVID-HL network, including existing and adapted validated scales and self-developed scales, was administered to 2996 University students in Florence. Gender differences were tested using the χ2 test or the Mann-Whitney U test

3) Male students reported a higher score in DHL than females (p<0.001). However, female students seek more often COVID-19 information on different sources (for themselves and other people), on various topics, consider various aspects of information quality to be “very important’’ (p<0.05) and are more likely to be “often dissatisfied’’ or ”partly satisfied’’ with information (p<0.001)

4)   Our study confirmed gender as an important dimension to explain students’ OHISB differences, which could help institutions to promote gender-specific education programmes and to provide gender-oriented health information.

The major weakness and strengths of the paper are as follows: 

1- comment regarding claim 1 w.r.t the following study:

A Guida, C Morittu, V Gallinoro, V Ferro Allodola, O Okan, K Dadaczynski, C Lorini, V Lastrucci, G Bonaccorsi, Digital Health Literacy during COVID-19: gender differences from a Florentine University experience, European Journal of Public Health, Volume 32, Issue Supplement_3, October 2022, ckac131.348, https://doi.org/10.1093/eurpub/ckac131.348

2- the claim of 2996 students' participation is not clearly shown in the data and results.

3- How the proposed scheme of study is compared with others. A comparative study/ table is missing to analyze the importance. 

4- however, the article is very well written and fairly structured, showing a good scheme of data display.

5. Para 2, 4 and 5 in the introduction section contain too much text which may be retracted.

Authors are required to accommodate the above comments to improve the overall contribution of the paper.

Reviewer 2 Report

This manuscript studied the gender differences in online health information-seeking behavior (OHISB) among college students during the COVID-19 pandemic. The data source for gender differences is based on questionnaires of Italian students between the first and second waves of the pandemic. The results show that the Digital Health Literacy (DHL) of male students are higher than that of females. This research confirms that gender is an important aspect of OHISB of students, which helps to promote gender-specific education programs in relevant institutions. This is a very meaningful work, which can provide ideas for the current work of researchers in this field. This manuscript can be published in Int. J. Environ. Res. Public Health, however, some revisions are needed before this article can be published.

1.      In introduction, the author says “for instance, “Social support for health” as one domain of the HLQ. In contrast, HL was higher in males in Chinese study.” But the definition of HLQ and HL is not clear, which may lead to the understanding deviations of readers, please give them a clear definition.

2.      In introduction, the author has too little description of the research purpose and research framework of this article. Please complete the part of the research purpose and research framework so that readers can have a better understanding of the research content.

3.      There are some formal mistakes in the full text, such as, on page 3, 2.2. “sociodemographic characteristics” should be “Sociodemographic characteristics”. Please check and correct the format issues.

4.      It is mentioned in the article that “Most 179 of the students (62%, N = 1862) attended a bachelor’s degree program, 37% (N = 1111) 180 attended a master's degree course, 1% a PhD, or post-doc program.” Will the proportion of gender differences appear different among people with different levels of education?

5.      On Page 10, the author said “causing higher levels of distress in women than in men.” What is the specific reason why women are more stressed?

6.      In introduction, the authors mentioned “Furthermore, it could have a positive impact on people in terms of adherence to treatment when they find adequate information, which is a dimension of individual Health Literacy” In order to support this statement, the following recently published important related papers should be cited: Adv. Mater. 2022, 34, 2106388; Sci. China: Chem. 2023, DOI: 10.1007/s11426-022-1477-x.

Round 2

Reviewer 1 Report

The authors have accommodated the suggestion.